# Disease Burden and Inpatient Management of Children with Acute Respiratory Viral Infections during the Pre-COVID Era in Germany: A Cost-of-Illness Study

**DOI:** 10.3390/v16040507

**Published:** 2024-03-26

**Authors:** Maren Alchikh, Tim O. F. Conrad, Patrick E. Obermeier, Xiaolin Ma, Brunhilde Schweiger, Onya Opota, Barbara A. Rath

**Affiliations:** 1Vaccine Safety Initiative, 10437 Berlin, Germany; maren.alchikh@vi-vi.org (M.A.); p.e.obermeier@gmail.com (P.E.O.); 2Laboratoire Chrono-Environnement, Université Bourgogne Franche-Comté, 25030 Besançon, France; 3ESGREV (ESCMID Respiratory Virus Study Group), 4001 Basel, Switzerland; onya.opota@chuv.ch; 4Zuse Institute Berlin, 14195 Berlin, Germany; conrad@zib.de; 5Department of Pulmonology, Capital Institute of Pediatrics, Beijing 100005, China; pediamaxl@outlook.com; 6Unit 17, Influenza and Other Respiratory Viruses, Department of Infectious Diseases, National Reference Centre for Influenza, Robert Koch-Institute, 13353 Berlin, Germany; bruni.schweiger@gmx.de; 7Institute of Microbiology, University of Lausanne, 1011 Lausanne, Switzerland

**Keywords:** influenza-like illness, respiratory virus, respiratory viral infection, direct medical cost, non-direct medical cost, social determinants of health

## Abstract

Respiratory viral infections (RVIs) are common reasons for healthcare consultations. The inpatient management of RVIs consumes significant resources. From 2009 to 2014, we assessed the costs of RVI management in 4776 hospitalized children aged 0–18 years participating in a quality improvement program, where all ILI patients underwent virologic testing at the National Reference Centre followed by detailed recording of their clinical course. The direct (medical or non-medical) and indirect costs of inpatient management outside the ICU (‘non-ICU’) versus management requiring ICU care (‘ICU’) added up to EUR 2767.14 (non-ICU) vs. EUR 29,941.71 (ICU) for influenza, EUR 2713.14 (non-ICU) vs. EUR 16,951.06 (ICU) for RSV infections, and EUR 2767.33 (non-ICU) vs. EUR 14,394.02 (ICU) for human rhinovirus (hRV) infections, respectively. Non-ICU inpatient costs were similar for all eight RVIs studied: influenza, RSV, hRV, adenovirus (hAdV), metapneumovirus (hMPV), parainfluenza virus (hPIV), bocavirus (hBoV), and seasonal coronavirus (hCoV) infections. ICU costs for influenza, however, exceeded all other RVIs. At the time of the study, influenza was the only RVI with antiviral treatment options available for children, but only 9.8% of influenza patients (non-ICU) and 1.5% of ICU patients with influenza received antivirals; only 2.9% were vaccinated. Future studies should investigate the economic impact of treatment and prevention of influenza, COVID-19, and RSV post vaccine introduction.

## 1. Introduction

The economic burden of influenza-like illness (ILI) associated with respiratory viruses (RVs) is poorly understood. RV infections (RVIs) are among the most common reasons for healthcare visits, especially in children [1,2]. The inpatient management of RVIs and ILI occupies significant personnel and institutional resources. During the COVID-19 pandemic, it became evident that RVIs may take clinics and hospitals to their limit. Some RVIs, such as those caused by SARS-CoV-2, respiratory syncytial virus (RSV), and influenza virus, have since become vaccine-preventable, and there are antiviral treatment options available for COVID-19 and influenza [3,4]. Additional antivirals against RVIs are in development [5,6], which will warrant adaptations in best practices and clinical decision making in the near future.

In this study, we make use of a unique setting at one of Europe’s largest pediatric academic hospitals to gain better understanding of the economic impact of the eight most common respiratory viral infections in children: influenza virus, RSV, human adenovirus (hAdV), human rhinovirus (hRV), human metapneumovirus (hMPV), human parainfluenza virus (hPIV), human bocavirus (hBoV), and human coronavirus (hCoV). We studied this question in the absence of universal pediatric vaccine recommendations for any of the eight RVIs.

The present study was conducted in the context of a six-year quality improvement (QI) program, where all patients aged 0–18 years who fulfilled a predefined ILI case definition (fever ≥ 38 °C and ≥1 respiratory symptom or physician-documented ILI) received laboratory RT-PCR diagnostics [7]. To determine disease severity and risk factors consistently across all patients participating in the QI program, we used a previously published and validated composite clinical score, the VIVI Disease Severity Score, capturing 22 key variables in real time via a mobile app (the VIVI ScoreApp, Vaccine Safety Initiative, Berlin, Germany) [7,8,9,10,11,12]. Disease severity and individual risk factors were therefore systematically captured in all patients in compliance with clinical data standards. In addition, the QI team recorded detailed clinical data throughout the hospital course such as standardized severity and risk scores and virologic testing, as well as clinical data including vaccination and treatment history. The QI program left the hospital routine unchanged; therefore, two systems ‘ran’ simultaneously. This program, therefore, was suitable to compare QI system data with the clinical decision making in routine care, such as the routine use of diagnostics, International Classification of Disease (ICD) coding, non-ICU admission, ICU admission, oxygen support, use of mechanical ventilation, virologic and imaging studies, and so forth.

Our team participated in a previously published international collaborative study that investigated the costs associated with RSV lower respiratory tract infection (LRTI) in hospitalized children [13]. In the present study, we expand the scope to include infections caused by seven other major RVs to put RSV costs into context i.e., which RVIs caused higher costs in the inpatient setting, and which RVIs most frequently lead to inpatient admissions or respiratory support measures. This question is particularly interesting after the global COVID-19 pandemic, when the costs of RVI management were widely discussed in the context of (a) the ethics of equity and fair allocation of resources, as well as (b) the impact of immunization, antiviral therapy, and other interventions in cost and disease severity.

In this study, we compare the costs of inpatient management and intensive care among children with laboratory-confirmed influenza, (seasonal) coronavirus infection, RSV infection, and infections with five other common RVs. We had the opportunity to study this question in a pre-(COVID-19) pandemic setting, at a time when influenza was the only RVI for which vaccines and/or antiviral therapy was available. There was no pediatric influenza vaccine recommendation in Germany [14], and antivirals were rarely used. Therefore, this study informs about the impact of ‘natural’ RVIs without the mitigating effect of vaccines and antiviral therapy. The decision to admit to the ICU is often due to a patient’s need for mechanical ventilation; therefore, we also recorded the requirement for mechanical ventilation.

Regarding the reimbursement of healthcare cost, Germany uses a modified version of the most widely used reimbursement system for inpatient health services, the diagnosis-related groups (DRGs) [15]. To make costs comparable with other countries, where no DRG system exists, and to assess the use of hospital resources in greater detail, we also calculated cost based on individual line items.

The medical profession strives for the equal treatment of all patients, but disparities still exist. Especially during the COVID-19 pandemic, equality became a concern, and emergency room triage systems and fair access to diagnostic testing, mechanical ventilation, intensive care, and antivirals were called into question.

Here, we studied, in a pre-pandemic cohort, the impact of social determinants of health (SDHs), i.e., surrogate markers captured in the context of the QI program, such as ethnicity, race, migratory background, birth rank, number of children, individuals in the household, and education). We investigated whether SDH can be linked to the likelihood of diagnostic procedures performed in the emergency department and/or the decision to admit to the hospital. We were interested in the question whether SDHs played any role in these decisions, or whether if there was equal access to virological testing, or admission to hospital (non-ICU or ICU)—solely dependent on the patient’s level of disease severity (as assessed by the VIVI ScoreApp) and/or individual risk.

The objectives therefore were as follows:(a)To identify the inpatient management costs associated with the eight most common RVIs in the pediatric age group.(b)To specify costs associated with hospitalization in a general ward versus ICU, and/or mechanical ventilation, continuous positive airway pressure (CPAP), or oxygen support.(c)To assess the relationship between risk factors, disease severity, and SDHs (using surrogate markers) regarding clinical decisions to perform diagnostic tests in the emergency department (ED), to admit to non-ICU/ICU wards, and to start mechanical ventilation/CPAP/oxygen support, respectively.

## 2. Materials and Methods

### 2.1. Cohort Analysis

As described previously [16,17], from December 2009 to March 2014, a specifically trained QI team performed standardized clinical assessments and virologic testing of all ILI patients aged 0 to 18 years at the point of care. The QI program was approved by the institutional review board (Charité EA 24/008/10). Informed consent procedures were waived for the purpose of enhanced quality of care and infection control.

Throughout the year, the QI team screened all hospitalized patients daily (including patients in the intensive care units) on admission as well as throughout their hospital stay, in addition to all patients presenting to the ED on Wednesdays, when private practices and GP Offices in Germany are usually closed. Patients with a documented physician diagnosis of ILI and/or fulfilling the QI ILI case definition automatically participated in the QI program. Influenza-like illness case criteria were defined as evidence of fever with a body temperature ≥38 °C and ≥1 respiratory symptom (including cough, rhinitis/coryza, red/sore throat, ear ache, dyspnea, tachypnea, labored breathing, and wheezing) [16]. Nasopharyngeal swabs were collected in a universal transport medium (CopanTM, Copan Diagnostics, Murrieta, CA, USA) and investigated at the National Reference Centre for Influenza at the Robert Koch-Institute, Berlin. Specimens were analyzed by real-time PCR for influenza A and B virus, RSV, hMPV, hAdV, hRV, hBoV (hBoV-1), hPIV (hPIV1-4), and hCoV (NL63, 229E, OC43) [9,18,19,20,21,22].

### 2.2. RT-PCR Analysis

Nucleic acid was extracted from nasopharyngeal swabs by a MagNA Pure 96 DNA and Viral NA Small Volume Kit (Roche, Basel, Switzerland), a MagAttract Viral RNA M48 Kit (Qiagen, Venlo, The Netherlands), or an RTP DNA/RNA Virus Mini Kit (Invitek, Berlin, Germany) according to the manufacturer’s instructions using a specimen volume of 200, 300, or 400 μL, respectively. Twenty-five microliters of extracted RNA were subjected to cDNA synthesis applying 200 U M-MLV Reverse Transcriptase (Invitrogen, Carlsbad, CA, USA) in a total reaction volume of 40 μL.

Specimens were analyzed for influenza A and B virus, RSV, hMPV, hAdV, and hRV by real-time PCR as published previously [8,17,18,19,20,21]. Investigation of hCoV (NL63, 229E, OC43, and HKU1), hPIV1-4, and hBoV-1 was performed in a total reaction volume of 15 μL containing 1× PCR buffer, 4 mM of MgCl_2_, 0.2 mM of dNTP with dUTP, 40 ng/μL of BSA, 0.3 U Platinum Taq Polymerase primers and probes, and 5 μL of cDNA (or nucleic acid for hBoV-1). Amplification was carried out at 95 °C for 300 s, followed by 45 cycles at 95 °C for 15 s and 60 °C for 30 s [9].

### 2.3. Patients’ Baseline Demographics

In this study, we examine relationships between variables in a contingency table by employing the chi-square test of independence. This function calculates the Pearson’s chi-squared statistic and corresponding *p*-value, which serve as indicators for testing the independence hypothesis among the observed frequencies in the contingency table. To compute the expected frequencies, we rely on the marginal sums under the assumption that the variables are independent [23].

To compute confidence intervals, we used the Wilson Score Interval method, which addresses the limitations of the normal approximation interval [24]. This technique introduces asymmetry into the process of interval estimation. The Wilson score interval, as opposed to the symmetric normal interval, avoids issues such as overshoot and zero-width intervals, making it a more robust option for analyzing small sample sizes and cases where the probability value is close to 0 or 1. This method of interval estimation is particularly useful for dealing with skewed observations and yields more precise results in statistical inference.

### 2.4. Analysis of Clinical Decision Making

Key indicators of clinical decision making assessed in the QI program included in the analysis were the same as above: (a) the performance of diagnostic tests (see below), (b) admission to non-ICU or ICU wards, and (c) the placement of a patient on oxygen, CPAP, or mechanical ventilation, respectively. Diagnostic testing included under (a) in our analysis involved chest X-ray and/or blood collection and/or performance of RV routine testing as per commercial multiplex PCR (Luminex xTAG^®^ RVP FAST V2; Luminex Corporation, Austin, TX, USA).

### 2.5. Cost Analysis

The primary outcome for this study was the cost of ILI-associated hospitalization from admission to discharge as well as ED costs for those ILI patients who remained outpatients, i.e., were discharged directly from the ED. Costs of inpatient stays were split into costs at the intensive care unit (‘ICU’, which included intermediate care units) versus non-ICU costs for the 8 different types of viral monoinfection. We compared 3 age groups: 0–5 years, 6–12 years, and 13–18 years.

We determined the total costs by combining the direct costs of patient care, including drug treatment cost, the cost of a hospital bed per day, nebulization, respiratory and physiotherapy, X-ray, consultation fees for physicians and affiliate health workers, and laboratory tests, as well as non-medical direct costs, including transportation costs, and non-medical indirect costs due to a loss of working hours by the caregiver.

Costs were then separately determined as per usual in Germany, i.e., using DRGs (with the state prime rates of 2015). DRGs are a classification system that assign patients to different ‘diagnosis-related groups’ that are clinically meaningful, with the goal to underwrite payment rates depending on the average effort and costs incurred for the management of patients with similar disease entities. To this end, DRGs are based on sociodemographic and diagnostic information and coded in line with the International Classification of Diseases and therapeutic information routinely collected during hospitalization [25]. Therapeutic information is coded with country-specific procedure coding systems, such as the Operation and Procedure Code used in Germany. In 2003, the German diagnosis-related groups were introduced for the reimbursement of inpatient treatment [26].

The analysis of costs had to be carried out within the constraints of the German system, i.e., two-fold: once according to actual costs (incurred due to services rendered), i.e., individual line items, and secondly, according to the billing code used in Germany, the so-called G-DRG system. The G-DRG system is based on average cost estimates per disease entity and is updated regularly using actual cost data from a representative sample of hospitals, as well as service data from all German hospitals. Cost data are determined uniformly for all participating hospitals using an actual cost approach on a full-cost basis according to a calculation manual. The calculation manual allows us to determine raw case costs (for the participating hospitals). These raw case costs form the basis for determining the German valuation ratios.

Individual cost assessments in this study were based on interviews by QI staff and a review of the Charité emolument agreement, wage agreement, and sales information for work medical equipment. We updated all costs to 2015 Euro/SU Dollar conversion rates using Bloomberg.com’s exchange rate of EUR 1 = USD 1.09, as of 10 December 2015.

Our study’s cost data were analyzed using the Two-Factor ANOVA (without replication) technique [27]. This statistical method offers a robust framework for examining the effects of two factors on a response variable, allowing for the evaluation of both the main effects of each factor and their interaction effects on the dependent variable of interest. By employing this technique, we gained a deeper understanding of the investigated factors and their effect on observed outcomes. The Two-Factor ANOVA without replication method provides a dependable method of statistical inference in situations characterized by limited resources.

### 2.6. Analysis of Clinical Decision Making in Relation to Risk-Adjusted Disease Severity (raVIVI Score) vs. Social Determinants of Health (SDHs)

For this analysis, we assessed clinical decisions in relation to the patient’s risk-adjusted severity score (raVIVI Score) which is calculated from the VIVI Disease Severity Score (‘VIVI Score’) and the VIVI Risk Factor Score as published previously: VIVI Disease Severity Score/(VIVI Risk Factor Score + 1) [10]. All patients were included in this analysis (*n* = 4776). The goal of the raVIVI Score is to simulate a clinician’s thinking, i.e., weighing the patient’s disease severity at the time of measurement, in relation to their individual risk of severe disease such as age or underlying conditions.

The raVIVI Score was measured with the help of the VIVI ScoreApp 1.0, a mobile application allowing healthcare professionals to grade disease severity and risk in their patients according to the criteria of the VIVI Score [28]. The VIVI Score is a previously published 22-item weighed clinical composite score, which was developed based on a systematic literature review and WHO criteria of uncomplicated and complicated disease [8]. The VIVI Score as well as the mobile app were subsequently validated in single- and multi-center studies in Europe (PEDSIDEA) and the USA [10]. VIVI Scores can range from 0 to 48 [8,10,11,12].

The VIVI Risk Factor Score captures underlying conditions that may lead to increased disease severity in patients with ILI and RVI, such as age < 2 or >65 years, chronic underlying conditions, and premature birth [8]. After assessing for risk factors unique to each patient, the VIVI ScoreApp asks for the 22 items of disease severity as per the VIVI Score, using terminologies and a data format fully compliant with CDISC standards [29].

For the purposes of this study, we assessed whether disease severity in relation to individual risk (i.e., the raVIVI Score) was the main driver of clinical decisions, as would be expected in a perfectly equitable and fair system.

However, while the raVIVI Score focuses on assessing disease severity in relation to individual risk factors, mirroring a clinician’s approach to evaluating patients, we also looked at alternative dimensions of clinical decision making: To explore the potential influence of social determinants of health (SDHs), we introduced the SDH Score. The SDH Score was implemented to investigate whether social factors, such as ethnicity, race, or education level, influence the decision-making process in healthcare. In essence, the SDH Score explores whether social determinants have an impact on clinical decision-making, revealing potential clinician bias. By analyzing both the raVIVI Score and the SDH Score, we aim to provide a comprehensive understanding of the factors that drive clinical decision making in healthcare settings.

The SDH Score was constructed using a set of seven patient characteristics: (i) ethnicity (other than not Hispanic or Latino); (ii) race (other than Caucasian or white); (iii) migratory background; (iv) birth rank (>1); (v) number of children (>2); (vi) number of individuals in the household (<3 or >3); and (vii) level of education (high school graduation prior to 10th grade). Each of these characteristics was assigned a binary value, with a score of one point for the presence of the characteristic and zero points for its absence or when the information was unknown.

In our statistical analysis, we employed logistic regression models to assess the relationship between the two scores, SDH and raVIVI, and binary outcomes, such as the decision for ICU admission or the choice to perform diagnostic tests [30]. The coefficients derived from these models provide insights into the extent of influence that each score has on clinical decisions. Additionally, we assessed the statistical significance of these coefficients through *p*-values which assess the evidence against the null hypothesis, which suggests no significant relationship between the scores and the clinical decision.

We used the logistic regression model to examine the relationship between SDH Score and ICU admission. The computed coefficients in this model provide insights into how a one-unit change in the SDH Score influenced the probability of ICU admission. In this context, the model’s intercept represents the log odds of ICU admission when the SDH Score is zero, serving as a baseline reference point. The odds ratio, on the other hand, quantifies the multiplicative change in the odds of ICU admission associated with a one-unit increase in the SDH Score. For instance, our findings indicate an odds ratio of 1.09, signifying that with each additional point in the SDH Score, there is a 9% higher likelihood of ICU admission, assuming that all other factors remain constant. This insight highlights the potential impact of social determinants of health on the decision to admit a patient to the ICU, beyond other variables considered in the analysis.

We conducted Random Forest analysis to assess the features of individual components comprising the SDH Score. This machine learning approach allowed us to explore the relative importance of individual determinants in predicting clinical outcomes. The analysis was performed using all features representing the elements of the SDH Score, the SDH Score itself, and the target variable representing a specific clinical outcome. We performed the Random Forest analysis with 100 estimators. The Random Forest model was trained on 80% of the data, with the remaining 20% used for testing. After training, we extracted the feature importance values provided by the model, which quantify the contribution of each feature to the prediction of the clinical outcome.

## 3. Results

### 3.1. Patients’ Baseline Demographics

From December 2009 to March 2014, 4776 pediatric ILI patients participated as a part of the QI program (55.8% male, median age of 1.6 years, age range of 0–18.8). Patients with influenza were older than the average RVI patients (median: 4.3 years; range: 0.1–18.8). Patients with RSV infection were younger on average (median: 0.8 years; range: 0.0–14.5) compared to patients with other types of RVI. There was a slight predominance of males among patients with hMPV and hBoV. Patients with hRV infection were slightly more likely to have underlying pulmonary conditions (11.4%, 95% CI: 9.1, 14.1), patients with hCoV infection were more likely to have underlying neurological conditions (9.2%, 95% CI: 4.5, 17.8) or to have been born prematurely (9.2%, 95% CI: 4.5, 17.8). Demographic characteristics and risk factors of patients with different RVIs are displayed in Table 1.

### 3.2. Analysis of RVIs in Relation to Clinical Decision Making

(a)Diagnostic Testing

We analyzed the proportion of patients receiving diagnostic tests (as outlined in 1.5. in the Methods Section) in relation to the RVI detected in the respective patient (Figure 1).

Human coronavirus infection (40.8%, 95% CI: 30.4, 52.0) followed by human parainfluenza virus infection (45.3%, 95% CI: 38.4, 52.4) were the rarest causes for diagnostic testing.

(b)Hospitalization and ICU admission

An analysis of the proportion of patients with different RVIs who required hospitalization and ICU admission is summarized in Figure 2.

Hospitalization: RSV infection was the primary reason for hospitalization in 77.2% (95% CI: 73.3, 80.7) of the patients; hBoV infection followed with 72.1% (95% CI: 66.2, 77.2) hospitalization.

ICU admission: For ICU admission, hPIV infection was the second most common cause (24.2%; 95% CI: 18.7, 30.8) and hMPV infection the third most common cause (23.7%; 95% CI: 17.2, 31.6).

(c)Mechanical Ventilation, CPAP, and Oxygen Supplementation

We analyzed the use of mechanical ventilation, CPAP, and O2 supplementation in relation to the RVI detected in the respective patient.

Mechanical ventilation was most used in patients with influenza (1%; 95% CI: 0.4, 2.5).

CPAP and O2 supplementation were most used in patients with RSVs, with CPAP at 1.8% (95% CI: 0.8, 3.4) and O2 supplementation at 33.3% (95% CI: 29.3, 37.6).

Table 2 shows the percentage of patients with specific RVIs who required mechanical ventilation, CPAP, and oxygen support.

### 3.3. Cost Analysis

We assessed the same clinical decisions (see Section 2.5) with regard to the cost of the respective line item vs. DRG fees in patients with different types of RVI. The cost showed a normal distribution.

As shown in Table 3, the mean costs for patients admitted to the ICU ranged from EUR 4260.40 for hAdV infection to EUR 29,261.31 for influenza in our setting. For patients in regular hospital wards (non-ICU), costs ranged from EUR 1498.38 for hAdV infection to EUR 1975.53 for hRV infection. Different age groups showed no significant differences, except for outpatients with influenza A and B virus infection, or infection with hAdV, hRV, hBoV, and hCoV. The costs between different RVIs were not significantly different, but the costs between ICU patients, non-ICU patients, and outpatients for the same RVI were significantly different for influenza and hBoV infection.

### 3.4. Clinical Decision Making in Relation to Risk-Adjusted Disease Severity (raVIVI Score) and Social Determinants of Health (SDHs)

For patients with similar levels of disease severity (VIVI Score) in relation to risk (VIVI Risk Factor Score), we investigated these as the main parameters linked to clinical decision making as outlined in Section 2.5.

#### 3.4.1. Risk-Adjusted Disease Severity Score (‘raVIVI Score’) vs. SDHs

As outlined in Section 2.6, we expressed disease severity in relation to individual risk with the risk-adjusted VIVI Disease Severity Score (raVIVI Score). We compared clinical decision making (as outlined in Section 2.5) in relation to the raVIVI Score versus SDHs.

(a)Diagnostic Testing

The coefficient for diagnostic testing compared to the raVIVI Score was 0.12 (95% CI: 0.10, 0.13), which means that, for an increasing raVIVI Score, the odds ratio increases by 12% per raVIVI Score point (*p*-value < 0.05).

(b)Non-ICU admission and ICU admission

For non-ICU admission, the coefficient was 0.10 (95% CI: 0.08, 0.11, *p* < 0.05) for the raVIVI Score, which means that, for an increasing raVIVI Score, the odds ratio increases by 10% per point. The coefficient was 0.13 (95% CI: 0.10, 0.17, *p*-value < 0.05) for the SDH score. The higher the SDH Score, the more the odds ratio increases (13% per SDH point).

For ICU admission, the coefficient for the raVIVI Score was 0.04 (95% CI: 0.03, 0.06, *p*-value < 0.05), with a 4% increase in the odds ratio per raVIVI Score point, whereas it was 0.09 (95% CI: 0.04, 0.13, *p*-value < 0.05) (9% increase in the odds ratio per point) for the SDH Score.

(c)Mechanical ventilation, CPAP, and O2 supplementation

For mechanical ventilation, the coefficient was −0.01 (95% CI: −0.06, 0.09) for the raVIVI Score, whereas the coefficient was −0.36 (95% CI: −0.79, −0.00) for the SDH Score (30% decrease in the odds ratio per SDH Score point). The coefficient for CPAP was −0.02 (95% CI: −0.08, 0.06) for the raVIVI Score compared to 0.11 (95% CI: −0.01, 0.38) for the SDH Score. The odds ratio for O2 supplementation increases 10% for every additional point of the raVIVI Score (coefficient: 0.10, 95% CI: 0.08, 0.11) and 19% for every additional point of the SDH Score (coefficient: 0.19, 95% CI: 0.14, 0.23).

The findings for the assessment of (c) versus the raVIVI Score and SDH are displayed in Table 4.

#### 3.4.2. Feature Importance Comprising the SDH Score

The SDH Score component consistently shows the highest feature importance across all clinical outcomes. Our findings are displayed in Table 5.

## 4. Discussion

### 4.1. RVI in Relation to Clinical Decision Making

We compared the costs of eight types of RVI in 2372 pediatric patients with RV mono-infections aged 0–18 in an academic hospital setting. This was performed in the unique context of a QI program, where a total of 4776 patients fulfilling the same ILI case definition were all tested for the eight most common RVs and monitored according to standard operating procedure. The QI program was run independently and in parallel to routine care, allowing for comparisons between the two systems. This unique inception cohort provided real-world data sufficient to model the actual costs elicited by different types of laboratory-confirmed RV monoinfections, with a known denominator.

In this pediatric hospital setting, the highest overall cost resulted from influenza, with EUR 2767.14 (non-ICU) and EUR 29,941.71 (ICU), followed by RSV infection with EUR 2713.14 (non-ICU) and EUR 16,951.06 (ICU). Among patients with RSV infection, 77.2% were hospitalized. Of them, 33.3% required oxygen supplementation and 31.5% were admitted to the ICU, compared to 13.2% with influenza.

Our analysis was focused on the cost of monoinfections to identify differences in economic impact for each individual RVI. For methodological reasons, our paper does not address viral–viral or viral–bacterial coinfections. This would be beyond the scope, as it would be impossible to discern the level of contribution of each individual co-infecting pathogen to the overall cost. Similarly, symptomatic cases where no pathogen was identified may be diverse in their pathogenesis and could not be summarized into a homogenous group.

While this is the most comprehensive comparison of 8 RVI to date, our data are well in line with trends observed by other groups assessing individual RVIs. A study by Al Amad et al. with 1811 patients (78% under 15 years of age) found an ICU admission rate of 23% in patients with influenza, 40% in patients with RSV and hAdV infection, 33% in patients with hMPV infection, and 42% in patients with hPIV infection [30]. These results are consistently higher than in our study. A retrospective Brazilian study with 12,160 children aged 0–12 years found a hospitalization rate of 47.6% (ICU admission rate: 4.8%) in patients with influenza, 59.5% (ICU: 9.5%) in patients with RSV infection, 64.3% (ICU: 14.3%) in patients with hAdV infection, and 63.8% (ICU: 4.8%) in patients with hPIV infection in 2009 [31].

Our cohort was unable to show the impact of vaccination against influenza, coronavirus, or RSV. Influenza immunization rates in our cohort were very low; RSV and coronavirus vaccines were not yet available at the time.

Immunization has proven to be one of the most effective interventions to prevent and control RVIs [32]. Most recently, vaccines have become available not only against influenza, but also RSVs and the pandemic coronavirus, SARS-CoV-2. No vaccines are available for other human coronaviruses, hRV, hMPV, hAdV, hBoV, or hPIV, whereas some are in (pre)clinical development. Live oral vaccines show promise in reducing the risk of respiratory hAdV infection and are in routine use in the United States military but are presently not available to civilians [33]. An hMPV vaccine is in development [34,35,36]. Attempts to produce a protective vaccine against hRV have failed due to large numbers of antigenically distinct serotypes and the lack of a suitable small-animal infection model to test candidate vaccines [37].

In Germany, influenza vaccination is only recommended for individuals above age 60 and/or people with underlying conditions. Coverage rates for influenza vaccination in Germany have been low even in these groups, with only 43.3% of the population aged 65 years and over in 2022 being immunized on average [38]. Small-scale policy interventions such as awareness campaigns have failed to increase influenza vaccine uptake [38,39]. Influenza vaccination is not generally recommended for healthy children in Germany, but even at-risk patients are often undervaccinated (around 16% age 15 years and younger) [40], possibly due to vaccine hesitancy [41]. In our setting, only 1.1% of healthy children and 5.8% of at-risk children were vaccinated during the respective season. In comparison the United States., with a universal influenza vaccine recommendation for all children 6 months of age and up, achieve influenza vaccine coverage rates around 50% in the pediatric age group [42]. Notably, our study was carried out prior to the COVID-19 pandemic and prior to the introduction of COVID-19 and RSV vaccines.

Our study shows the usefulness of innovative digital tools in helping to capture critical data at the point of care. With the VIVI ScoreApp, we had a quick and reliable way to classify disease severity and risk, which comes in handy, especially in the ED. The goal of the raVIVI Score is to reflect/approximate clinical thinking, i.e., the clinician asking themselves the question, ‘How ill is this patient in relation to their individual risk?’ The use of automated risk-adjusted severity scoring could help to speed up the triage of patients with ILI and RVIs during outbreaks and pandemics [43].

### 4.2. Cost Analysis

In our economic analyses, we used the individual mono-RVI case numbers (*n* = 2372) as the foundation for analysis. The analysis of costs had to be carried out within the constraints of the German system.

Most studies undertaken to date have set out to derive a national-level cost estimate of the impact of a specific disease, based on some version of the cost-of-illness approach, which was first formalized by Rice and colleagues in the late 1960s [44]. Using this approach, the possible economic consequences of specific illnesses are divided into ‘direct costs’, i.e., the expenses incurred because of the illness itself (including medical care, travel costs, etc.), and ‘indirect costs’ such as the value of lost productivity due to sick leave. Direct and indirect costs are then summed up to provide the overall societal costs of an illness. According to WHO guidance, economic impact studies raise multifaceted health policy considerations at both the macroeconomic level (society) and the microeconomic level (households, firms, governments) [45]. Economic burden studies help to identify possible strategies for reducing the cost of disease through preventive or treatment strategies.

Cost-of-illness studies are important for the evaluation of healthcare systems. Analyzing and comparing costs can be beneficial for stakeholders and policymakers. These analyses are significant because RVIs, such as influenza, COVID-19, and RSV, are widespread, taking up significant resources. Prior to the COVID-19 pandemic, it was estimated that RSV infections cause approximately 33 million acute LRTIs worldwide, including 3.6 million hospital admissions and 26,300 deaths in children below 5 years of age [46]. The global direct medical costs for the management of RSV LRTIs in children under 5 years of age was at EUR 4.82 billion [13].

Because virological testing is not carried out universally in patients with ILI [7,47], estimating the economic burden associated with specific RVIs remains a challenge. Our analysis demonstrated that influenza incurred the highest cost in patients requiring admission to the ICU, followed by infections with RSV, hRV, hMPV, hCoV, hBoV, hPIV, and hAdV. The costs associated with influenza infection were EUR 29,261.31 for direct and non-direct medical costs and EUR 680.40 for indirect costs per ICU stay per patient. High costs for the management of influenza infection were caused by the longer duration of mechanical ventilation as well as the larger rate of patients requiring mechanical ventilation (average rate: 1% with a 12.25-day duration of mechanical ventilation for influenza vs. a rate of 0.41% and 10.5 days for RSV infection). This discrepancy might be due to the wide-spread use of high-flow oxygen and CPAP at our center. High-flow O2 and non-invasive ventilation such as CPAP did not lead to high costs because they do not count as a ventilation bed during billing.

The published literature comparing the costs associated with RVIs in children in different regions is sparse. Data from low–middle-income countries (LMICs) are even more challenging to come by. A study in Kenya with 275 patients of all ages defined USD 117.86 among inpatients with influenza and USD 19.82 for outpatients [48]. Zhang et al. presented a systematic review and meta-analysis of RSV-associated costs in children [13]. In 41 studies reporting data from 1987 to 2017, the average cost per episode was EUR 3452 and EUR 299 for inpatient and outpatient management, respectively, which compares well to our findings of EUR 3584 and EUR 85 derived from the DRG system. In a retrospective study conducted in the United States involving 815 children < 18 years with hMPV, the hospital cost per patient was USD 5513 [49].

Reimbursement policies may impact clinical decision making. For example, in Germany, there is usually only a small per capita flat rate reimbursed for the treatment of a patient in the ED, regardless which services were provided. Also, most test results are not ready at the time of the ED discharge to the home or to a hospital ward. Diagnostic tests for RVIs (including rapid tests for RSVs, influenza virus, and other RVs) are therefore often deferred and reserved for inpatients, where such tests will be reimbursed at a better rate. The disadvantage of such policies is that, for example, children may be sent home with, e.g., acute influenza, possibly infecting others, including immunocompromised individuals or elderly people [50]. The implications and societal costs may be considerable. With more vaccines and specific antiviral treatment options becoming available, and diagnostic tests more sensitive and easier to perform, rapid-turnaround diagnostics should be used more broadly in emergency departments, allowing for timely infection control and treatment [51]. Ever since the SARS-CoV-2 pandemic, rapid diagnostics have been introduced into the workflow, at least temporarily. During the second phase of the COVID-19 pandemic from June 2022 to February 2023, every citizen of Germany was entitled to at least one free rapid SARS-CoV-2 antigen test per week. Patient advocacy organizations such as Families Fighting Flu in the United States [52] increasingly favor virological testing to be performed in the emergency department so that RVIs can be treated at the time when antivirals are most effective. Another benefit may be that unnecessary prescriptions of antibiotics can be saved. In the future, hospital workflow and triage protocols may need to be revised in this regard. Here, health insurers and politicians must create the conditions to implement change in reimbursement and best practices.

In the G-DRG System, diagnostic tests performed in inpatient units are not reimbursed separately in the per-case fee (except for SARS-CoV-2 tests), which means that more expensive tests may not be performed routinely (such as viral culture, sequencing, resistance testing, individual PCR, or next-generation sequencing for the detection of rare or emerging viruses).

We previously reported from the same QI program that even among hospitalized symptomatic ILI patients, only 8.7% ever had any virus diagnostics done in routine care [7]. This means that RVI diagnoses are being missed in routine care—which means that the specific RV detected will also not be ICD-coded and reimbursed via the DRG system. Thus, the introduction of DRGs represented a fundamental reorientation of hospital remuneration systems replacing the formerly prevailing principle of self-cost recovery, in which hospitals were able to include individual reasons for certain cost structures in their budget agreements [25]. In other words, diagnosis-based reimbursement systems may be of questionable use from an infectious disease perspective, as billing does not fully reflect the actual effort, or the timeliness of actions taken.

The present work has several limitations. We analyzed the costs of one center in a high-income country during a specific time frame. No conclusions can be drawn from our data about the costs of RVIs in other countries, particularly low-resource settings. This QI program was undertaken prior to the COVID-19 pandemic; therefore, no results are available on SARS-CoV2. However, we did study pre-pandemic coronaviruses in this paper. Similar studies may need to be carried out in the future, comparing this novel virus to the ones that were in circulation prior to the COVID-19 pandemic.

We were able to study costs and medical decision making regarding specific RVIs in a unique setting that was highly suitable to explore this question. Several factors in a patient–doctor encounter will influence clinical management. Cultural stereotypes may not be conscious, but these and other factors may influence how information is processed, potentially leading to unintended implicit bias in decision making. Research suggests that implicit bias may contribute to healthcare disparities along the lines of race, ethnicity, gender, and other characteristics [53,54].

### 4.3. Clinical Decision Making in Relation to Risk-Adjusted Disease Severity (raVIVI Score) and Social Determinants of Health (SDHs)

In an impartial setting where no bias in the management of RVIs exists, treatment would entirely depend on disease severity and individual risk. Patients would only receive diagnostic tests or be admitted to inpatient units based on comparable levels of disease severity in relation to risk. In a setting where bias does exist, social determinants of health (SDHs) would influence medical decision making such as ICU/non-ICU admission, treatment, or the duration of hospital stay [53,54,55]. In our study, we show that admission to the ICU versus a (non-ICU) regular ward depended not only on a patient’s individual disease severity/risk (raVIVI Score), but also on SDHs. The SDH Score allowed us to look for potential bias in clinical decision making. Like the raVIVI Score, higher SDH Scores correlated with an increased likelihood of ICU admission, which could indicate a differential approach in managing patients with varied social backgrounds.

The introduction of the SDH Score in this context constitutes an innovative approach to exploring the potential influence of social factors on healthcare delivery. We recognized the importance of validating the assumption of consistent effect size per point change in the scores. To address this, we conducted a series of sensitivity analyses, including the introduction of polynomial terms in our logistic regression models. These analyses allowed us to detect and account for any non-linear relationships between the scores and outcomes. Our findings confirmed that while there were some variations, the general trend of the relationship remained consistent. SDHs are not routinely captured in medical care. The QI program did capture a significant number of SDH elements proactively but left it up to the participant to reply on a voluntary basis. This is usual practice in handling sensitive data. The SDH Score developed by the QI team is a practical way to introduce capturing SDH data in busy clinical settings. On average, 50–60% of SDH questions were answered. To air on the side of caution, the SDH Score measures only positively reported SDH elements. This may lead to an underestimation of SDH effects due to underreporting. To provide a maximum level of data standardization, all VIVI ScoreApp and SDH data are computed according to CDISC (www.cdisc.org, accessed on 23 October 2023) data standards and terminologies, which are used to capture interoperable data for clinical trials and observational studies. This analysis not only supports the relevance of the SDH Score in clinical decision making but also provides insights into the underlying structure of the SDH determinants in relation to health outcomes. This reinforces the validity of the SDH Score as a meaningful composite measure of social determinants impacting health.

Effective triage in busy ED settings may help to streamline the workflow, improving equity and the overall quality of patient services [56]. Triage means not only ranking in terms of importance (prioritization) but also the just allocation of limited resources. Survival and quality of life are often viewed in relation to the use of resources [57]. Accurate and effective triage not only saves lives but also furthers the fair allocation of resources [58]. The VIVI ScoreApp (https://immunisationhubs.eu/projects/technology-platform/vivi-scoreapp/, accessed on 16 January 2024) is the only ILI/RVI-specific score that is independent of the type of RVI, can be used in children, can be used during triage as well as follow-up, and has been validated in tens of thousands of patients by now. This severity score needs to be differentiated from predictive scores, such as the SOFA score (which is based on survival) [59].

The recent COVID-19 pandemic has raised concerns over ageism and a lack of equity in accessing resources [60]. Nacoti et al. who worked at a hospital in Bergamo heavily impacted by the COVID-19 pandemic, reported that the care provided at centralized hospitals was not sufficiently patient-centered and should be complemented or replaced by more community-centered care. In times of crisis, medical centers may be forced to operate below their usual standard of care. ICU beds may be sparse and access to palliative care diminished. Families may lose touch with relatives or be diverted to other institutions [61]. The authors suggested a three-level procedure: level 1: mild or moderate cases that can be treated at home; level 2: severe cases receiving care in community centers; and level 3: critical cases admitted to the hospital [62]. The VIVI ScoreApp might allow physicians to utilize an objective measure of risk-adjusted disease severity at the point of care, obtained via the mobile app within 1–2 min, to categorize levels of urgency [8]. In cases of suspected ILI/RVI, the VIVI ScoreApp may complement the generic Manchester Triage System in a simple and meaningful way, supporting equity and quality of care at the same time.

## 5. Conclusions

Our study illustrated that the major costs for ICU hospitalization were caused by influenza virus, while RSVs led to most ICU admissions. RSVs also led to most CPAP and O_2_ supplementation. Use of diagnostics and hospitalization depended not only on severity and risk factors but also on SDH.

More studies on the costs of viral respiratory infections, especially during epidemics and pandemics, could support policymakers in making decisions about the allocation of resources, investments in vaccine development and preventive as well as therapeutic interventions. The VIVI ScoreApp could work as an equity tool. It could help to make the decisions in EDs fairer.

We conclude that the standardization of risk and severity assessments as well as the proactive assessment of SDH (e.g., via the VIVI ScoreApp) may ultimately contribute to a greater understanding of equity in patient management before, during, and after epi/pandemics. Future studies will continue to monitor costs during the introduction of new vaccines against COVID-19 and RSV infections. Equal access to vaccine prevention and antiviral treatment matters and should support quality of care and cost-effectiveness.

## Figures and Tables

**Figure 1 viruses-16-00507-f001:**
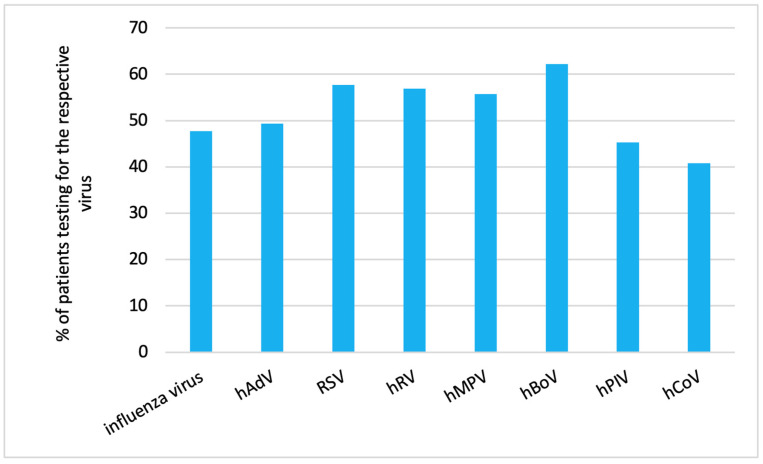
Proportion of patients with different RVIs who required diagnostic testing.

**Figure 2 viruses-16-00507-f002:**
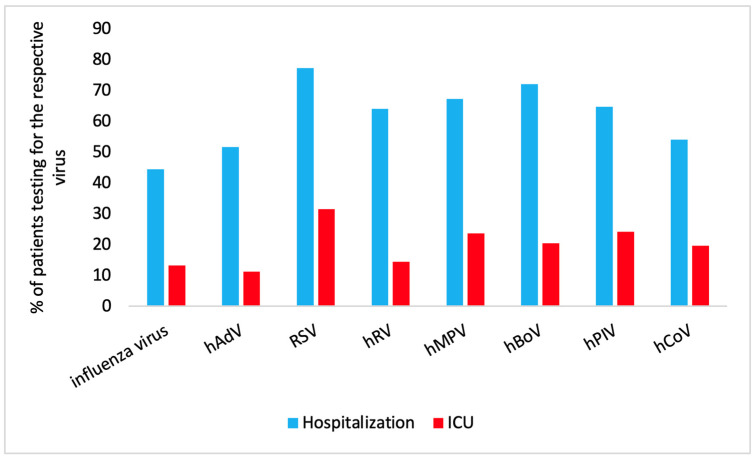
Proportion of patients with different RVIs who required hospitalization (blue) and ICU admission (red).

**Table 1 viruses-16-00507-t001:** Demographics characteristics and underlying risk factors for all ILI patients, patients with different RVIs, RV co-infections, and no RV detected.

	All Patients	Influenza Virus	hAdV	RSV	hRV	hMPV	hBoV	hPIV	hCoV	**Co-Infection**	**No Virus Detected**	***p*-Value ***
*n* (% of total)	4776 (100)	409 (8.6)	203 (4.3)	492 (10.3)	617 (12.9)	131 (2.7)	254 (5.3)	190 (4.0)	76 (1.6)	969 (20.3)	1435 (30.1)	
Age (years) Median [IQR]	1.6 [0.7–3.7]	4.3 [1.6–7.9]	1.8 [0.9–2.5]	0.8 [0.3–1.8]	1.6 [0.7–2.8]	1.6 [0.7–2.8]	1.5 [0.8–2.5]	1.4 [0.6–2.6]	1.2 [0.6–7.9]	1.3 [0.6–2.4]	2.1 [0.9–6.1]	0.50
Gender Male (%)	55.8	54.5	57.1	56.7	60.1	60.3	58.3	51.6	61.8	55.8	53.3	0.79
Chronic condition Pulmonary (95% CI)	8.1% (7.4–8.9)	7.3% (5.2–10.3)	2.5% (1.1–5.6)	8.5% (6.4–11.3)	11.4% (9.1–14.1)	8.4% (4.8–14.4)	10.6% (7.4–15.0)	10.0% (6.5–15.1)	7.9% (3.7–16.2)	7.4% (5.9–9.3)	7.3% (6.1–8.8)	0.20
Cardiac (95% CI)	7.5% (6.7–8.2)	5.9% (4.0–8.6)	4.4% (2.4–8.2)	6.9% (5.0–9.5)	8.8% (6.8–11.2)	6.9% (3.7–12.5)	6.7% (4.2–10.5)	9.5% (6.1–14.5)	5.3% (2.1–12.8)	5.9% (4.6–7.6)	9.1% (7.7–10.7)	0.54
Metabolic (95% CI)	3.5% (3.0–4.1)	5.6% (3.8–8.3)	3.0% (1.4–6.3)	2.4% (1.4–4.2)	4.4% (3.0–6.3)	2.3% (0.8–6.5)	2.8% (1.3–5.6)	4.7% (2.5–8.8)	2.6% (0.7–9.1)	2.5% (1.7–3.7)	4.9% (3.9–6.1)	0.42
Hepatorenal (95% CI)	2.7% (2.3–3.3)	1.2% (0.5–2.8)	3.0% (1.4–6.3)	2.6% (1.6–4.5)	3.7% (2.5–5.5)	1.5% (0.4–5.4)	2.0% (0.8–4.5)	2.6% (1.1–6.0)	2.6% (0.7–9.1)	2.4% (1.6–3.5)	0.5% (0.2–1.0)	<0.05
Neurological (95% CI)	5.0% (4.4–5.6)	5.4% (3.6–8.0)	2.0% (0.8–5.0)	3.3% (2.0–5.2)	4.9% (3.4–6.9)	5.3% (2.6–10.6)	3.9% (2.2–7.1)	4.2% (2.2–8.1)	9.2% (4.5–17.8)	4.5% (3.4–6.0)	6.3% (5.1–7.7)	0.56
Haemato-oncological/immunological (95% CI)	2.4% (2.0–2.9)	2.7% (1.5–4.8)	1.5% (0.5–4.3)	1.6% (0.8–3.2)	2.6% (1.6–4.2)	3.8% (1.6–8.6)	2.4% (1.1–5.1)	2.6% (1.1–6.0)	1.3% (0.2–7.1)	1.9% (1.2–2.9)	0.2% (0.1–0.6)	<0.05
Prematurity < 33 weeks GA (95% CI)	5.2% (4.6–5.9)	4.2% (2.6–6.6)	3.0% (1.4–6.3)	4.7% (3.1–6.9)	6.0% (4.4–8.2)	6.9% (3.7–12.5)	5.1% (3.0–8.6)	5.8% (3.3–10.1)	9.2% (4.5–17.8)	4.2% (3.1–5.7)	5.9% (4.8–7.3)	0.90
Any (95% CI)	24.4% (23.2–25.7)	19.3% (15.8–23.4)	13.8% (9.7–19.2)	12.4% (9.8–15.6)	15.7% (13.1–18.8)	22.1% (15.9–30.0)	12.6% (9.1–17.2)	19.5% (14.5–25.7)	15.7% (9.3–25.6)	16.3% (14.1–18.8)	19.8% (17.8–21.9)	0.10

* *p*-value: if <0.05 then there is difference in the groups.

**Table 2 viruses-16-00507-t002:** Percentage of patients with specific RVIs who required oxygen support, CPAP, and mechanical ventilation.

	O2 % of Total (95% CI)	CPAP % of Total (95% CI)	Mechanical Ventilation % of Total (95% CI)
influenza virus	7.8 (5.6;10.8)	0.7 (0.3;2.1)	1.0 (0.4;2.5)
hAdV	3.5 (1.7;7.0)	0.0 (0.0;1.9)	0.0 (0.0;1.9)
RSV	33.3 (29.3;37.6)	1.8 (1.0;3.4)	0.4 (0.1;1.5)
hRV	21.1 (18.0;24.5)	0.7 (0.3;1.7)	0.3 (0.1;1.2)
hMPV	24.4 (17.9;32.4)	0.8 (0.1;4.2)	0.0 (0.0;2.9)
hBoV	15.4 (11.4;20.3)	0.0 (0.1;1.5)	0.4 (0.1;2.2)
hPIV	12.1 (8.2;17.5)	0.5 (0.1;2.9)	0.0 (0.0;2.0)
hCoV	4.0 (1.4;11.0)	1.3 (0.2;7.1)	0.0 (0.0;4.8)
*p*-value *	<0.05	0.82	0.98

* Statistically significant if *p*-value < 0.05.

**Table 3 viruses-16-00507-t003:** Total cost per episode for DRGs and individual line items in EUR (direct and non-direct medical costs) and indirect costs per episode for patients with different laboratory confirmed RVIs.

Total *n* = 2372			Diagnosis-Related Groups (DRGs) (Direct + Non-Direct Medical Cost) − Total Cost per Episode (EUR)	Summary of Individual Items (Direct + Non-Direct Medical Cost) − Total Cost per Episode (EUR)	Indirect Cost − Total Cost per Episode (EUR)	Total of Individual Items (Direct + Non-Direct Medical Cost) and Indirect Cost (EUR)
Influenza virus *n* = 409						
	ICU *n* = 54		7854.24	29,261.31 *	680.40	29,941.71
		0–5 years *n* = 32	5624.56	4262.32	680.40	4942.72
		6–12 years *n* = 16	7943.44	30,632.53	907.20	31,539.73
		13–18 years *n* = 6	3511.87	1500.98	1360.80	2861.78
	Non-ICU *n* = 128		1668.35	1973.34 *	793.80	2767.14
		0–5 years *n* = 88	1761.25	1507.83	793.80	2301.63
		6–12 years *n* = 19	1569.35	2419.47	992.25	3411.72
		13–18 years *n* = 21	1533.18	1483.31	595.35	2078.66
	Outpatient *n* = 227		85.00	88.33 *	340.20	428.53
		0–5 years *n* = 131	85.00	95.17	340.20 **	435.37
		6–12 years *n* = 65	85.00	88.33	340.20 **	428.53
		13–18 years *n* = 31	85.00	87.03	340.20 **	427.23
hAdV *n* = 203						
	ICU *n* = 23		3881.00	4260.40	680.40	4940.80
		0–5 years *n* = 22	3893.12	3471.61	567.00	4038.61
		6–12 years *n* = 0	NA	NA	NA	NA
		13–18 years *n* = 1	3582.43	4247.96	680.40	4928.36
	Non-ICU *n* = 82		1622.04	1498.38	595.35	2093.73
		0–5 years *n* = 79	1611.92	1498.38	595.35	2093.73
		6–12 years *n* = 2	1784.26	2357.88	992.25	3350.13
		13–18 years *n* = 1	1581.10	2357.88	1984.50	4342.38
	Outpatient *n* = 98		85.00	87.03	340.20	427.23
		0–5 years *n* = 89	85.00	87.03	340.20 **	427.23
		6–12 years *n* = 8	85.00	60.58	340.20 **	400.78
		13–18 years *n* = 1	85.00	60.58	340.20 **	400.78
RSV *n* = 492						
	ICU *n* = 155		6487.58	15,817.06	1134.00	16,951.06
		0–5 years *n* = 152	6356.71	14,436.10	907.20	15,343.30
		6–12 years *n* = 1	7144.93	24,895.13	4082.40	28,977.53
		13–18 years *n* = 2	5581.03	15,223.48	2494.80	17,718.28
	Non-ICU *n* = 225		3584.00	1973.34	739.80	2713.14
		0–5 years *n* = 223	3781.44	1973.65	793.80	2767.45
		6–12 years *n* = 1	3544.82	1457.72	595.35	2053.07
		13–18 years *n* = 1	3132.45	2848.09	1190.70	4038.79
	Outpatient *n* = 112		85.00	87.57	340.20	427.77
		0–5 years *n* = 108	85.00	95.17	340.20	435.37
		6–12 years *n* = 4	85.00	86.81	340.20	427.01
		13–18 years *n* = 0	NA	NA	NA	NA
hRV *n* = 617						
	ICU *n* = 89		6451.92	13,486.82	907.20	14,394.02
		0–5 years *n* = 76	6684.73	13,486.31	907.20	14,393.51
		6–12 years *n* = 13	6253.43	4269.72	680.40	4950.12
		13–18 years *n* = 0	NA	NA	NA	NA
	Non-ICU *n* = 306		1792.65	1973.53	793.80	2767.33
		0–5 years *n* = 276	1852.98	1973.51	793.80	2767.31
		6–12 years *n* = 24	1791.21	2438.78	992.25	3431.03
		13–18 years *n* = 6	1782.43	1939.73	793.80	2733.53
	Outpatient *n* = 222		85.00	87.57	340.20	427.77
		0–5 years *n* = 185	85.00	88.33	340.20 **	428.53
		6–12 years *n* = 30	85.00	87.57	340.20 **	427.77
		13–18 years *n* = 7	85.00	60.58	340.20 **	400.78
hMPV *n* = 131						
	ICU *n* = 31		4259.31	5653.70	907.20	5653.70
		0–5 years *n* = 31	4259.31	5653.70	907.20	6560.90
		6–12 years *n* = 0	NA	NA	NA	NA
		13–18 years *n* = 0	NA	NA	NA	NA
	Non-ICU *n* = 57		2133.87	1508.05	595.35	2103.40
		0–5 years *n* = 53	2384.78	1508.01	595.35	2103.36
		6–12 years *n* = 4	2044.68	1711.53	694.58	2406.11
		13–18 years *n* = 0	NA	NA	NA	NA
	Outpatient *n* = 43		85.00	87.57	340.20	427.77
		0–5 years *n* = 38	85.00	87.57	340.20	427.77
		6–12 years *n* = 5	85.00	86.81	340.20	427.01
		13–18 years *n* = 0	NA	NA	NA	NA
hBoV *n* = 254						
	ICU *n* = 52		4231.89	5863.96 *	680.40	6544.36
		0–5 years *n* = 46	4287.90	5863.96	680.40	6544.36
		6–12 years *n* = 4	4256.98	9067.01	1474.20	5728.92
		13–18 years *n* = 2	4178.61	4254.72	680.40	9747.41
	Non-ICU *n* = 131		2287.66	1973.40 *	793.80	2767.20
		0–5 years *n* = 114	2383.77	1507.98	595.35	2103.33
		6–12 years *n* = 14	2159.62	1945.65	793.80	2739.45
		13–18 years *n* = 3	2256.30	4209.63	1786.05	5995.68
	Outpatient *n* = 71		85.00	87.57 *	340.20	427.77
		0–5 years *n* = 66	85.00	87.57	340.20 **	427.77
		6–12 years *n* = 3	85.00	60.58	340.20 **	400.78
		13–18 years *n* = 2	85.00	60.58	340.20 **	400.74
hPIV *n* = 190						
	ICU *n* = 46		3883.92	4274.76	680.40	4955.16
		0–5 years *n* = 43	3973.11	4274.76	680.40	4955.16
		6–12 years *n* = 0	NA	NA	NA	NA
		13–18 years *n* = 3	3631.74	4247.12	680.40	4927.52
	Non-ICU *n* = 77		1799.54	1973.14	793.80	2766.94
		0–5 years *n* = 70	1832.85	1973.07	793.80	2766.87
		6–12 years *n* = 7	1746.22	1938.67	793.80	2732.47
		13–18 years *n* = 0	NA	NA	NA	NA
	Outpatient *n* = 67		85.00	87.57	340.20	427.77
		0–5 years *n* = 64	85.00	87.57	340.20	427.77
		6–12 years *n* = 3	85.00	41.38	340.20	381.58
		13–18 years *n* = 0	NA	NA	NA	NA
hCoV *n* = 76						
	ICU *n* = 15		4894.56	5644.87	907.20	6552.07
		0–5 years *n* = 14	4174.00	5632.71	907.20	6539.91
		6–12 years *n* = 0	NA	NA	NA	NA
		13–18 years *n* = 1	6422.98	8370.77	1360.80	9731.57
	Non-ICU *n* = 26		2464.31	1498.42	595.35	2093.77
		0–5 years *n* = 23	2581.74	1498.42	595.35	2093.77
		6–12 years *n* = 2	2478.73	3072.61	1289.93	4362.54
		13–18 years *n* = 1	2347.91	3734.62	1587.60	5322.22
	Outpatient *n* = 35		85.00	88.33	340.20	428.53
		0–5 years *n* = 29	85.00	87.57	340.20 **	427.77
		6–12 years *n* = 5	85.00	60.58	340.20 **	400.78
		13–18 years *n* = 1	85.00	31.85	340.20 **	372.05

* *p* < 0.05 significant difference in costs between ICU admission/non-ICU admission/outpatients of the viruses. ** *p* < 0.05 significant difference in costs between the age groups. Groups with *n* = 0 were not included.

**Table 4 viruses-16-00507-t004:** Clinical decision making depending on raVIVI Score and SDH.

Predictor	Outcome	Coefficient	95% CI	Odds Ratio	*p*-Value
raVIVI Score	Diagnostic Test	0.12	0.10, 0.13	1.12	6.76^−60^
raVIVI Score	Non-ICU admission	0.10	0.08, 0.11	1.10	4.23^−49^
raVIVI Score	ICU admission	0.04	0.03, 0.06	1.04	8.30^−11^
raVIVI Score	O2 Supplementation	0.10	0.08, 0.11	1.11	9.90^−43^
raVIVI Score	CPAP	−0.02	−0.08, 0.06	0.98	0.76
raVIVI Score	Mechanical ventilation	−0.01	−0.06, 0.09	0.99	0.71
SDH Score	Diagnostic Test	−0.02	−0.07, 0.01	0.98	0.08
SDH Score	Non-ICU admission	0.13	0.10, 0.17	1.14	1.86^−11^
SDH Score	ICU admission	0.09	0.04, 0.13	1.09	3.47^−4^
SDH Score	O2 Supplementation	0.19	0.14, 0.23	1.21	5.01^−14^
SDH Score	CPAP	0.11	−0.01, 0.38	1.12	0.07
SDH Score	Mechanical ventilation	−0.36	−0.79, −0.00	0.70	4.84^−2^

**Table 5 viruses-16-00507-t005:** Predictive feature importance (ethnicity, race, birth rank, number of children, individuals in the household, and level of education) comprising the SDH Score.

Clinical Outcome	Ethnicity	Race	Migratory Background	Birth Rank	Number of Children	Individuals in the Household	Level of Education	SDH Score
Diagnostic Test	0.08	0.08	0.09	0.15	0.09	0.14	0.07	0.31
Non-ICU admission	0.07	0.09	0.10	0.12	0.10	0.17	0.05	0.30
ICU admission	0.09	0.10	0.10	0.11	0.09	0.10	0.08	0.33
O2 Supplementation	0.07	0.07	0.09	0.13	0.13	0.15	0.05	0.31
CPAP	0.11	0.11	0.06	0.14	0.11	0.14	0.00	0.32
Mechanical ventilation	0.01	0.27	0.13	0.08	0.03	0.16	0.00	0.32

## Data Availability

Data are contained within the article.

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
