# Peer review of "Disease Burden and Inpatient Management of Children with Acute Respiratory Viral Infections during the Pre-COVID Era in Germany: A Cost-of-Illness Study"

_viruses, 2024, doi:10.3390/v16040507_

Round 1
Reviewer 1 Report
Comments and Suggestions for Authors
COMMENTS
Thank you for the opportunity to review the manuscript. The authors examined clinical decision making and costs associated with respiratory viral infections in a pediatric population.
Major Points:
Both the items used and scoring scheme for the Social Determinants of Health (SDH) score can easily be perceived as subjective unless the authors can cite prior validated research. Were any missing data observed among the 7-items, and if so, how was that handled?
The 2nd most common group after no-virus are those with co-infections (n=969; 20.3%). How was co-infection defined for this study? Was it a co-infection among the 8 viruses of interest or something else? This detail is essential so the reader can understand, for example, cost for all patients with influenza detections vs. patients with influenza detection only. Relatedly, when presenting the cost comparisons and decision making it doesn’t appear the co-infection groups are combined (e.g., RSV-alone + RSV/hCoV) but wanted to make sure.
The summary results specify mean cost. Did cost exhibit a roughly normal distribution? Otherwise, consider either transforming the data or utilizing non-parametric options.
The raVIVI is used to depict disease severity, so how do we interpret increased raVIVI had decreased odds of hospitalization, yet increased odds of ICU admission? Similarly, higher SDH is decreased odds for hospitalization and increased odds for ICU admission. Relatedly is a 1-point change in raVIVI a meaningful unit change, considering the range is [0,48]? Lastly, this makes the assumption that the increased odds of the outcome was constant with each 1-point change. Were the data checked to see if that assumption was valid?
Minor Points:
Was the ED screened only once per week (line 145)? If so, would that introduce a limitation in terms of representativeness? Additionally, if only once per week, was that day of the week allowed to vary or was it the same for all 5 years?
The stated implication of the p-value (line #268) is false. Additionally, the designation of p-values simply as <0.05 and >0.05 (Table 3) is not appropriate. There’s little reason why the actual numbers could not be provided.
Consider providing the IQR rather than absolute range (e.g., age in Table 1).
Were any costs and/or decision making comparisons done when compared with no virus detected?
The labeling of the figures and figure titles could be improved. For example, presumably the ‘hospitalization’ bar and ‘ICU’ bars in Figure 2 are not mutually exclusive, yet the figure title states ‘hospitalization vs. ICU’.
Would strongly recommend turning Figure 3 into a table to better appreciate CPAP and MVent utilization.
Were adjusted logistic models completed where raVIVI and SDH were both included? This would help with the claim that ‘’hospitalization and ICU admission depend not only from a risk factor but also from SDH” [line 524]
Comments on the Quality of English Language
a couple of very minor typos noted.
Reviewer 2 Report
Comments and Suggestions for Authors
The authors have discussed a very important and hard to analyse issue regarding the medical costs of different etiological entities in acute respiratory viral infection. Their study pointed out important findings regarding the different direct and indirect costs of medical care for the studied viruses but also how their costs are modified due to age category. Also they have studied the different per age admission rate in the ICU of their population pointing out the importance of vaccination not only with pneumococcal vaccine but also with influenza vaccine
Author Response
Thank you for your report.
Reviewer 3 Report
Comments and Suggestions for Authors
The introduction,methods ,results and conclusion are well written.The only minor mistake is in line 322-word regards.
Author Response
Thank you for your comment. Please see the attachment (reviewer 1).
Round 2
Reviewer 1 Report
Comments and Suggestions for Authors
I thank the authors for addressing reviewer comments and modifying the text.
Major Points:
The clarification on the SDH Score is insufficient to demonstrate the validity of the intended latent construct. The authors could consider completing a factor analysis or machine learning approach (e.g., random forest) to show the score validity. An alternative would be to examine the 7 questions separately. In either case, careful consideration is recommended with handling the non-insignificant amount of missing data (“on average 50-60% of SDH question were answered”) and whether single imputing is sufficient.
Per the authors, only single-virus detections were included for the cost comparison, implying the co-infection (N=969) and no virus (N=1435) groups were excluded. This should leave a total of 2372 records. Presumably only these 2372 records were used when examining clinical decision making (Table 3) but that’s unclear. The authors could consider having co-infection and no virus be part of the inclusion/exclusion criteria and have a fixed denominator throughout.
Having hospitalization be a primary outcome is still confusing since only hospitalized patients are screened six of out every seven days. Dichotomizing the data into hospitalized-non ICU vs. hospitalized ICU is more intuitive to the reader and does not introduce confusion by having an outcome and cohort inclusion be the same.
Comments on the Quality of English Language
none
Round 3
Reviewer 1 Report
Comments and Suggestions for Authors
I thank the authors for the efforts in responding to reviewer comments.